# The Effect of Resistance Training on Bone Mineral Density in Older Adults: A Systematic Review and Meta-Analysis

**DOI:** 10.3390/healthcare10061129

**Published:** 2022-06-17

**Authors:** Danilo A. Massini, Flávio H. Nedog, Thiago P. de Oliveira, Tiago A. F. Almeida, Caroline A. A. Santana, Cassiano M. Neiva, Anderson G. Macedo, Eliane A. Castro, Mário C. Espada, Fernando J. Santos, Dalton M. Pessôa Filho

**Affiliations:** 1Graduate Programme in Human Development and Technology, São Paulo State University (UNESP), Rio Claro 13506-900, SP, Brazil; dmassini@hotmail.com (D.A.M.); airthiago@yahoo.com.br (T.P.d.O.); tiagofalmeida.w@gmail.com (T.A.F.A.); caroline.santana@uemg.br (C.A.A.S.); merussi.neiva@unesp.br (C.M.N.); andersongmacedo@yahoo.com.br (A.G.M.); eliane.castro@unesp.br (E.A.C.); dalton.pessoa-filho@unesp.br (D.M.P.F.); 2Department of Physical Education, University Center of São Paulo State (UNICEP), Rio Claro 13500-200, SP, Brazil; henriquenedog@gmail.com; 3Department of Physical Education, Claretiano University Center, Rio Claro 13506-257, SP, Brazil; 4Department of Physical Education, São Paulo State University (UNESP), Bauru 17033-360, SP, Brazil; 5Faculty of Physiotherapy, Minas Gerais State University (UEMG), Divinópolis 35501-170, MG, Brazil; 6Laboratory of Exercise Physiology Research Group (LFE—Research Group), Universidad Politécnica de Madrid (UPM), 28040 Madrid, Spain; 7Polytechnic Institute of Setúbal, Department of Science and Technology (ESE-CIEF, EST-CDP2T), 2914-504 Setúbal, Portugal; fernando.santos@ese.ips.pt; 8Life Quality Research Centre (CIEQV-Leiria), 2040-413 Rio Maior, Portugal; 9Faculty of Human Kinetics, University of Lisbon, 1499-002 Lisboa, Portugal

**Keywords:** bone mineral content, strength training, aging

## Abstract

Resistance training (RT) has been considered an intervention with effective stimulus on bone mineral formation and is, therefore, recommended to decrease the rate of bone morpho-functional proprieties loss with aging. Thus, this meta-analysis aimed to analyze the effectiveness of RT protocols in promoting changes in bone mineral density (BMD) in older adults. The systematic reviews and meta-analysis followed the PRISMA guidelines (PROSPERO CRD42020170859). The searches were performed in the electronic databases using descriptors according to the PICO strategy. The methodological quality and risk of bias were assessed with the PEDro scale, and the magnitude of the results was determined by Hedges’ g. Seven studies involving 370 elderlies, with the RT planned as a unique exercise mode of intervention, showed designs with four to five exercises for upper- and lower-limbs musculature, two to three sets per exercise, eight to twelve repetitions to failure at 70–90% 1 RM, 60–120 s of rest between sets, and executed three times per week for 12–52 weeks. The RT protocols were classified between good and excellent and evidenced a positive effect on the BMD at the hip (0.64%) and spine (0.62%) but not in the femoral neck (−0.22%) regardless of the intervention length. The narrow range of either positive or negative changes in the BMD after the RT intervention support, at best, a preventive effect against the increasing risk of bone frailty in an older population, which is evident beyond 12 weeks of RT practice engagement.

## 1. Introduction

Bone mineral density (BMD) refers to the standardized bone mineral content per unit of area [1]. Thus, it reflects the integrity of bone tissue, being an indication of structural remodeling capacity [2], and, therefore, when reduced, it is an index of the risk of frailty and pathologies, such as osteopenia and osteoporosis, that have a negative impact on health [3,4,5]. Osteoporosis affects about 30% of all postmenopausal women in Europe and the United States, and it is estimated that 40% of these women will suffer one or more osteoporotic fracture in their lifetime [6]. As with other tissues, bone mass is subject to the aging process, with declining morphological and functional proprieties due to biological factors (muscle mass reduction and neuromuscular activation) and lifestyle (sedentariness and poor eating habits) [7,8,9]. For example, BMD reductions can reach rates of 0.6, 1.1, and 2.1% for the age groups 60–69, 70–79, and ≥80 years, respectively [10]. However, strength exercise can avoid BMD reduction from 1 to 3% per year of life when compared to adults not regularly engaged in exercise programs [11], highlighting that resistance exercise training (RT) offers adequate mechanical stress to stimulate BMD improvements in specific anatomical locations, such as the femoral neck (FN), total hip (TH), and lumbar spine (LS), where its decrease could be a problem [4,12,13].

Weightlifting exercises, such as resistance training (RT), have also been used as a strategy to prevent or mitigate BMD loss. This type of exercise induces tensional stimuli to the bones, which produce mechanical modulation of the osteogenic response and, hence, increase the bone mineral content [14,15]. Nevertheless, aerobic training and other forms of low-impact exercise, such as cycling and swimming, have been shown to have little or no effect on bone health [6,12]. It is also important to note that multi-joint exercises have clinical contributions to bone mineralization in older individuals. For example, long-term RT protocols (4–6 months), with moderate to heavy loads (50–80% of one repetition to maximum, 1 RM), two to three sets for exercise, and three sessions for weeks, including squats and deadlifts exercises, have demonstrated to improve the BMD of the spine and hip, as well as in the upper limbs, until 3.8%, which is considered clinically relevant [5]. Indeed, free weight exercises performed with high loads (>70% 1 RM) suggest the greatest enhancement of the BMD [14,16].

Specifically, regarding the effect of RT on the BMD in older adults, the desirable osteogenic effect has been considered preventive since minimal measurable changes in bone mass or/and density are supported by tissue metabolism improvement [13,17,18,19]. In fact, a considerable number of longitudinal studies have reinforced that RT is the favorable exercise mode to enhance bone health simply by avoiding loss in BMD values over time, or it evidences small increases, with a protocol from 3 to 12 months of the interventional period [4,13], despite other studies reporting small reductions in BMD with RT [17,20] and despite confounding factors, such as age, sex, ethnicity, nutritional status, conditioning level, bone mineral health, different experimental designs (duration, exercises, and others), perhaps having influenced the lack of responsiveness among the studies.

In particular, the aforementioned discrepancy in the results reported for different RT studies for older adults makes it difficult to provide evidence of the effectiveness of BMD stimulation and thus limits the assertiveness regarding the optimal RT planning (volume, intensity, and rest) for maintaining bone health in older adults. Therefore, the present systematic review aimed to analyze the effect of RT protocols on the BMD of specific anatomical locations (FN, TH, and LS) susceptible to osteoporosis in the older population, examining whether different volume and intensity settings could differentially affect the BMD at the anatomical locations considered.

## 2. Materials and Methods

This systematic review and meta-analysis of experimental studies with resistance training intervention followed the recommendations of the Preferred Reporting Items for Systematic Reviews and Meta-Analyses (PRISMA) [21] (see Appendix A). The high-sensitivity search was elaborate, using descriptors according to population, intervention, comparison, outcome, study design (PICOS) strategy: Populations: “Elderly” AND “Health”; Intervention: “Resistance Exercise” OR “Resistance Training”; Comparator: pre-post difference in the BMD as a result of an RT program (the comparison with the control group was not performed because the studies used comparisons with different types of exercises, e.g., impact, vibration, or aerobic exercise, or did not perform exercises, and this would influence the estimate of the effect size [22]); Outcomes: “Bone Density” OR “Bone Mineral Content” OR “Bone Remodeling” OR “Bone Metabolism”; and Study design: any study that included an RT intervention in older adults (aged 55+ years and with a body mass index below 30 kg/m^2^) with relevant outcomes for BMD assessed pre- and post-training was considered for inclusion. The reliability of the search strategy was verified by finding the study by Bemben et al. [17]. After that, computer searches were performed in the electronic databases: PubMed, ScienceDirect, and Scielo for studies published until 20 December 2020. In addition, citation tracking of the included studies was carried out through the databases Pubmed, Scopus, and Google Scholar. Gray literature (e.g., abstracts, conference proceedings, editorials, dissertations, and thesis) was not included [22]. The selected articles’ references were also searched to add relevant titles. Finally, attempts were made to contact the authors of the selected articles via e-mail to request any lack of relevant information. In order to avoid any selection bias, searches were performed by two authors (D.A.M. and T.P.O.). After the searches were conducted, the authors compared the lists of included and excluded studies; the discrepancies observed were analyzed through discussion and agreement with a third author (D.M.P.F.). This systematic review has received the registration number CRD42020170859 in PROSPERO baseline records.

### 2.1. Study Selection Criteria

Studies were included that showed quantification of BMD. The inclusion criteria adopted were: (i) complete studies performed in humans over 55 years old (to ensure selection of studies including only participants with a recognized trend of age-related effects on BMD) [23,24]; (ii) only studies that used dual-energy X-ray absorptiometry (DXA) as the assessment tool, a validated BMD measurement technique [25]; (iii) peer-reviewed and published in English; and (iv) in the last 10 years. The exclusion criteria adopted were: (i) studies conducted in clinical populations that interfere with bone metabolism (i.e., diabetics, obese) [3]; (ii) with average body mass index ≥ 30.0 kg/m^2^; (iii) studies that combined other types of exercises (i.e., vibration, impact, and aerobic exercises); (iv) studies that administered supplements or osteogenic drugs; (v) case studies and literature reviews (systematic review and meta-analysis).

### 2.2. Data Extraction

The data were extracted by two authors (F.H.N. and T.P.O.) using a pre-pilot spreadsheet and independently verified by a third author (D.A.M.) from the review team. The following data were extracted: (i) authors’ names, (ii) year of publication, (iii) population characteristics (sample size, sex, age, height, and body mass), (iv) characteristics of the resistance training protocol (series, repetitions, intensity, rest interval, weekly frequency, and intervention duration), (v) BMD measures (region and pre- and post-training values).

### 2.3. Assessment of Methodological Quality and Risk of Bias

This procedure was performed by two authors (D.A.M. and F.H.N.) using the 11-point Physiotherapy Evidence Database (PEDro) scale, which gives the study 1 point if the criterion is met or 0 if not [26]. The discrepancies observed were analyzed by a third author (D.M.P.F.). As criterion 1 concerns external validity, it was considered in the total score; similarly, criteria 5, 6, and 7 were removed due to the impossibility in physical exercise intervention studies to allocate groups of participants blindly; in addition, researchers rarely act blindly [27]. With the removal of these items, the maximum value on the PEDro scale is 7 points, with adjusted scores ranging from 0–3 being “poor quality”, 4 being “moderate quality”, 5 being “good quality”, and 6–7 being “excellent quality” (Table 1) [27,28].

### 2.4. Statistical Analysis

The magnitude of the study results was determined by Hedges’ g and 95% confidence interval (CI_95%_) due to the small sample size (n < 30) of some included studies [29]. For these estimates, the sample size, mean values, and standard deviation of the BMD pre- and post-training for each anatomical location (FN, TH, and LS) were used for each condition (RT protocols applied) of each study included in the meta-analysis. The relative effects of the training (∆%) were provided in percentages according to Equation (1).
(1)$$\Delta \%=\left[\frac{\left({\bar{x}}_{post}-{\bar{x}}_{pre}\right)}{{\bar{x}}_{pre}}\right]\cdot 100$$
where “∆%” is the effect of training in percent, “$\bar{x}$*_pre_*” is the average of the BMD before training, and “$\bar{x}$*_post_*” is the average of the BMD after training. Inconsistency was checked using the meta-analysis results and was based on visual inspection of Hedges’ g estimates, whether CI_95%_ overlapped or not, and also on statistical tests for heterogeneity determined by combining Cochran’s Q test with Higgins’ test [30] and its value categorized as: 0 < *I*^2^ ≤ 25%, indicating non-heterogeneity; 25% < *I*^2^ ≤ 50%, indicating low heterogeneity; 50% < *I*^2^ ≤ 75%, indicating moderate heterogeneity; and >75%, indicating high heterogeneity between studies [31]. Fixed-effect model was employed in the absence of inconsistency (*I*^2^ ≤ 25%). Publication bias analyses (Egger’s test) were not evaluated due to the inferiority of ten studies included [32,33,34]. Effect sizes for Hedges’ g were defined as: <0.20 = trivial, 0.20–0.50 = small, 0.50–0.80 = moderate, >0.80 = large [34,35]. Finally, meta-regression analyses were performed to ascertain the relationship between the interventional period (weeks) and the effect size (Hedges’ g) of the training for each anatomical location. The meta-analysis and statistical analysis were conducted using Comprehensive Meta-Analysis, version 2.0 (Biostat; Englewood, NJ, USA). A significance level of α = 0.05 was adopted for all statistical procedures.

## 3. Results

Figure 1 depicts the flow chart for all the systematic review and meta-analysis steps.

The mean percentage of change for the BMD of the selected articles was: the FN = 0.22 ± 0.64% (CI_95%_ = −0.59–0.16%), TH = 0.64 ± 0.51% (CI_95%_ = 0.32–0.95%), and LS = 0.62 ± 0.81% (CI_95%_ = 0.12–1.12%). Table 1 presents the main characteristics of the seven included studies [4,12,13,17,20,36,37]. These studies were selected independently by three reviewers, being published in European countries (57%) and the United States of America (43%). This totaled the inclusion of 370 participants, 62% female (61.9 ± 5.0 years) and 38% male (69.0 ± 9.0 years).

**Table 1 healthcare-10-01129-t001:** Main characteristics of the selected studies concerning the population features, RT protocol, effects on BMD, and quality analysis.

Study	Participants					Resistance Training Protocols							Bone Mineral Density (g/cm^2^)				PEDro
	n	Age	Height	Body Mass	BMI	Exercise	Sets	Rep	Intensity	Rest	Weekly Frequency	Duration	Bone	Interventions			
		(years)	(cm)	(kg)	(kg/m^2^)	(Body Region)			(%1 RM)	(s)		(Weeks)	(Region)	Pre	Post	∆ (%)	Scores
Mosti et al., 2013 [4]	8 W	61.9 ± 5.0	169.3 ± 6.5	72.3 ± 7.7	25.3 ± 2.9	1 ex. LL (SM)	24	8–12 3–5	50 90	120 180	3	12	FN LS TH	0.651 ± 0.084 0.759 ± 0.061 0.751 ± 0.125	0.655 ± 0.088 0.762 ± 0.067 0.756 ± 0.123	0.61 0.40 0.67	6 Excellent
Marques et al., 2011 [12]	23 W	67.3 ± 5.2			28.8 ± 4.6	4 ex. LL (LP, KE, LC, HAb) 4 ex. T and UL (CP, LR, SP, AC)	3	6–8	75–80	120	3	32	FN TH	0.684 ± 0.082 0.859 ± 0.124	0.676 ± 0.090 0.873 ± 0.132	−1.17 1.63	5 Good
Whiterford et al., 2010 [13]	73 M	64.6 ± 6.0	176.6 ± 6.8	82.4 ± 10.7	26.4 ± 3.1	4 ex. LL (HF, HE, Hab, CR) 5 ex. T and UL (WC, WE, BC, TP, FPS)	3	10	85	60	3	52	FN LS TH	0.966 ± 0.142 1.236 ± 0.189 1.045 ± 0.161	0.969 ± 0.140 1.235 ± 0.185 1.054 ± 0.157	0.31 −0.08 0.86	6 Excellent
Bemben et al., 2010 [17]	22 W	64 ± 0.9	160.6 ± 1.7	76.6 ± 3.1		5 ex. LL (LP, HF, HE, HAb, HAd) 3 ex. T and UL (SP, LPD, SR)	3	10	80	60	3	32	FN LS TH	0.902 ± 0.021 1.163 ± 0.028 0.955 ± 0.025	0.898 ± 0.021 1.156 ± 0.030 0.952 ± 0.024	−0.44 −0.60 −0.44	5 Good
Bocalini et al., 2009 [20]	40 W	69 ± 9.0	*≅155	68 ± 6.0	28 ± 4.0	5 ex. LL (LP, LC, KE, HAb, HAd) 7 ex. UL (CP, LPD, BC, TP, SR, SP, AC)	3	10	85	60	3	24	FN LS	0.705 ± 0.001 0.881 ± 0.001	0.704 ± 0.001 0.880 ± 0.001	−0.14 −0.14	6 Excellent
Marques et al., 2013 [36]	23 M 24 W	68.2 ± 5.2 68.2 ± 5.7	*≅169 *≅150	83.0 ± 11.7 64.2 ± 10.2	29.2 ± 3.4 28.6 ± 4.1	4 ex. LL (LP, KE, LC, HAb) 4 ex. T and UL (CP, LR, SP, AC)	3	6–8	75–80	150	3	32	FN LS TH FN LS TH	0.822 ± 0.113 1.051 ± 0.161 1.004 ± 0.140 0.715 ± 0.119 0.877 ± 0.122 0.864 ± 0.108	0.821 ± 0.115 1.065 ± 0.172 1.006 ± 0.138 0.705 ± 0.104 0.896 ± 0.129 0.872 ± 0.111	0.12 1.33 0.20 −1.40 2.17 0.93	6 Excellent
Bemben & Bemben 2011 [37]	45 M 79 W	65.2 ± 0.5 63.8 ± 0.4	176.8 ± 1.0 163.2 ± 0.6	83.5 ± 1.6 69.7 ± 1.5	26.7 ± 0.5 26.1 ± 0.5	7 ex. LL (KF, KE, LP, HF, HE, HAb, HAd) 5 ex. T and UL (FF, FE, SP, LPD, SR)	3 3 3 3	8 16 8 16	80 40 80 40		2 2 3 3	40	FN LS TH FN LS TH FN LS TH FN LS TH	0.903 ± 0.020 1.143 ± 0.035 0.943 ± 0.022 0.902 ± 0.019 1.186 ± 0.032 0.940 ± 0.019 0.894 ± 0.022 0.889 ± 0.021 0.950 ± 0.026 0.928 ± 0.026 1.186 ± 0.031 0.976 ± 0.031	0.902 ± 0.020 1.155 ± 0.034 0.949 ± 0.022 0.904 ± 0.019 1.195 ± 0.034 0.943 ± 0.019 0.889 ± 0.021 1.190 ± 0.034 0.956 ± 0.025 0.932 ± 0.026 1.190 ± 0.031 0.984 ± 0.031	−0.11 1.05 0.64 0.22 0.76 0.32 −0.56 0.96 0.63 0.43 0.34 0.82	5 Good

*: Data entered by the authors; RT: resistance training; BMD: bone mineral density; M: men; W: women; BMI: body mass index; FN: femoral neck; TH: total hip; LS: lumbar spine.; UL: upper limbs; LL: lower limbs; T: trunk; HI: high intensity; LI: low intensity: s: seconds; rep: repetitions. AC: abdominal curl; BC: biceps curl; CP: chest press; CR: calf raise; FE: forearm extension; FF: forearm flexion; FPS: forearm pronation/supination; HAb: hip abduction; HAd: hip adduction; HE: hip extension; HF: hip flexion; KE: knee extension; KF: knee flexion; LC: leg curl; LP: leg press; LPD: lateral pull-down; LR: lateral raise; SM: squat movement; SP: shoulder press; SR: seated row; TP: triceps pushdown; WC: wrist curls; WE: wrist extension

Improvements in BMD were evidenced but without statistical support in four of the eleven training protocols analyzed in the FN (Figure 2A), nine out of ten in the TH (Figure 2B), and seven out of ten in the LS (Figure 2C). The values of the methodological quality presented by the PEDro scale were good to excellent, with a range of five to six points.

The meta-analysis was performed from a fixed model for all anatomical locations because they did not present inconsistency (<25%) for the BMD of the FN (*I^2^* = 0.00%; Q_10_ = 7.998; *p* = 0.629), TH (*I^2^* = 0.00%; Q_9_ = 1.848; *p* = 0.994), and LS (*I^2^* = 22.78%; Q_9_ = 11.656; *p* = 0.23) (Figure 2A–C, respectively), as well as no significant changes for the BMD of the FN (*g* = 0.059, CI_95%_ = −0.214–0.096, *p* = 0.456 [trivial]), TH (*g* = 0.115, CI_95%_ = −0.044–0.274, *p* = 0.155 [trivial]), and LS (*g* = 0.064, CI_95%_ = −0.097–0.226, *p* = 0.437 [trivial]), which characterizes RT as an exercise with no significant effect on BMD in the older adults. The meta-regression results also showed no significant effect of the RT period on the BMD of the FN, TH, and LS (Figure 3A–C, respectively), demonstrating that the period of the RT protocols (12–52 weeks) did not significantly alter the BMD either positively or negatively.

## 4. Discussion

The systematic search selected seven studies with eleven different RT protocols planned for older people of both genders. The meta-analysis showed no effect of the RT on the bone density program for any of the anatomical locations: FN, TH, and LS; the meta-regression also demonstrated no additional effect of the length of interventions on BMD changes. We found no evidence that the BMD of the older adults increases significantly with resistance training alone.

Despite bone tissue not offering enough plasticity compared to other tissues [38,39], Going and Laudermilk [40] state that RT increases BMD by about 1 to 3% in the FN and LS anatomical locations for both premenopausal and postmenopausal women. These authors stated that minimal measurable increases are expected for the population, having no aging effect on the peak of bone mass beyond that achieved while young, and that individuals over 55 years presented larger increases in BMD due to bone susceptibility to decline with aging. Other studies did not corroborate the above-mentioned statement and reported only low changes in the BMD of the FN, TH, and LS, which ranged from 0.1 to 2.0% in participants > 60 years old engaged in RT protocols [24,41].

Most BMD values for the FN, TH, and LS changed positively after RT protocols lasting from 12 to 52 weeks but with no statistical difference, while non-training control individuals changed negatively accordingly, as expected, reinforcing that the effect of RT on BMD is to prevent natural reductions or to slow the rate of natural decrease with aging [13]. In fact, after revising the effect of exercise training programs on bone mass, Wolfe et al. [42] indicated the rate of bone mass loss prevention or reversion attained ≅1% per year for the FN and LS anatomical locations in older women who exercised regularly. Furthermore, the study of Rhodes et al. [18], with women aged 65 to 75 years engaged in an RT program for 54 weeks with a high-intensity load (75% 1 RM), reported small, although not significant, improvements in the FN (0.82 ± 0.11 vs. 0.83 ± 0.12 g/cm^2^), Ward’s triangle (W) (0.69 ± 0.13 vs. 0.70 ± 0.11 g/cm^2^), greater trochanter (0.74 ± 0.10 vs. 0.75 ± 0.11 g/cm^2^), and LS (1.10 ± 0.17 vs. 1.13 ± 0.18 g/cm^2^), while the control group reduced, also non-significantly, or had unchanged BMD for the FN (0.78 ± 0.09 vs. 0.73 ± 0.10 g/cm^2^), W (0.63 ± 0.10 vs. 0.59 ± 0.12 g/cm^2^), greater trochanter (0.69 ± 0.12 vs. 0.67 ± 0.11 g/cm^2^), and LS (1.01 ± 0.17 vs. 1.01 ± 0.17 g/cm^2^). Another study reporting the effect of RT on BMD involved postmenopausal women training with a moderate-intensity load (60 to 70% 1 RM) for 24 weeks [19]. The authors observed no changes in the BMD for the LS (0.01%) and FN (0.04%) in the resistive-trained group, while the control group reduced the BMD for the LS (−0.89%) and FN (−1.54%).

On the other hand, Zhao et al. [15] found no significant effect of resistance-alone training protocols for postmenopausal women on the BMD for the FN (0.21 g/cm^2^, CI_95%_ = −0.04–0.47 g/cm^2^) and LS (0.18 g/cm^2^, CI_95%_ = −0.10–0.46 g/cm^2^), but, rather, the authors considered the effect satisfactory at least to avoid reductions in BMD during the period of intervention. However, when an exercise program includes RT in combination with high-impact or weight-bearing exercises (tennis, running, dancing), the authors observed higher improvements in the BMD for the FN (0.41 g/cm^2^, CI_95%_ = 0.18–0.64 g/cm^2^) and LS (0.43 g/cm^2^, CI_95%_ = 0.16–0.70 g/cm^2^). This aligns with previous research and reinforces the idea that RT ensures bone health by maintaining the BMD in older adults. However, conflicting results regarding BMD alterations are reported among studies [4,11,17,40], which can be attributed to the differences in training overload, such as volume (two to four sets, three to fifteen repetitions) and intensity (80–90% 1 RM or 8–10 RM), training period (2.5–12 months), participants’ characteristics (postmenopausal women or men), sample size (21–143 participants), and nutritional control (e.g., replacement therapy or estrogen deficient) [4,13,20,43]. The impacts of the characteristics and prescription of RT programs on the changes in BMD are discussed below.

The effect of load intensity on the BMD of each anatomical location was reported by Maddalozzo and Snow [44], analyzing RT protocols with high- (70–90% 1 RM) and moderate-intensity loads (40–60% 1 RM). The authors reported, in men, significant increases in the BMD of the LS, trochanter, and whole body with the high-intensity protocol and increments also for the trochanter with the moderate-intensity training. Women only showed differences (considered in percent terms) in the BMD of the trochanter and FN with the high-intensity protocol, with no difference for moderate-intensity training. The differences between anatomical locations among men could be explained by the intraosseous osteogenic stress thresholds of response [4,44,45]. Women might require greater stimulation to obtain better BMD responses, although the current data are insufficient for any statement, and some authors argue that increased mechanical stimuli (volume or intensity) do not translate into additional increases in BMD [37].

Regarding the number of exercises in RT protocols, the studies have usually planned four to five exercise types for muscles in the lower limbs (e.g., hip and knee extensors and flexors, calf, hip adductors, and abductors), upper limbs (elbow flexors and extensors, and shoulder flexors and abductors), and trunk (pectoral, back, and abdomen) performed both on machines and free weight [12,13,20,37,46], which will act in different ways at the anatomical locations, with regional and whole-body effects [3,5]. However, the number of exercises planned for different body regions did not determine the magnitude of the effect on BMD [4,5,17,36]. Probably, larger changes associated with RT protocols are accounted for by the mechanosensiblity of bone tissue, which seems to be high in individuals with lower values of BMD, as is the case in osteopenia and/or osteoporosis [36,37].

For most of the RT protocols, three sets have been planned [13,17,47] despite studies comparing the effect of set number on BMD reporting that two sets are satisfactory to ensure positive results, mainly when combined with four or five exercises for muscles of different body regions [3,5]. Furthermore, the number of repetitions usually planned ranged from eight to twelve, with the load intensity at 60 to 85% 1 RM [20,36]. Finally, the rest duration observed between sets ranged from 60 to 120 s [3,5], and the frequency of the training sessions per week was close to three [12,17,20].

The analysis from meta-regression (Figure 3) suggested that the duration of the RT protocols did not influence the values of BMD beyond 12 weeks of training. No significant increases in BMD are observed in periods between 17 and 52 weeks [18,19], which supports the statement that the effect of RT on the bone mass in an older population is to prevent age-related reductions [13,48]. Theoretically, a longer RT protocol period is supported by the statement that the process of bone formation, resorption, and mineralization requires 3 to 4 months, and stabilization of bone mass content/density occurs in approximately 6 to 8 months [23]. However, the results of this review did not support the theory that longer interventions might have a greater impact on BMD since only one of the studies included in the present analyses performed a training protocol shorter than six months and the duration of RT did not influence the BMD response.

Although RT can positively influence bone remodeling, the biochemical process responsible for this effect is not yet fully understood [20,49]. However, other factors may mediate the effects of physical training on bone health, such as the individual’s nutritional status, genetics, and hormonal homeostasis [17], which explain why some studies have shown no positive effects of RT on BMD [2]. The main limitation of the present study is that the included studies did not present criteria for determining the sample size, which makes it impossible to confirm that the sample numbers were appropriate to identify the proposed outcome. Therefore, future studies should analyze the effect of RT protocols (with and without combinations of impact exercises) on the bone remodeling capability of an older population at an increased risk of tissue mineral disturbance and functional frailty, as well as verify gender differences.

## 5. Conclusions

Collectively, the reviewed studies show that RT has neither a positive nor a negative effect on the bone mineral integrity in a healthy older population of both genders. In addition, the results showed no greater effects on the BMD with longer interventions. From the revision of the RT protocols, it was possible to observe that RT should be planned with four to five exercises for the trunk, upper limbs, and lower limbs, containing two to three sets per exercise, eight to twelve repetitions at 70 to 90% 1 RM, rest ranging from 60 to 120 s between sets, and three sessions per week for 12–52 weeks. However, these general guidelines seem better linked with the prevention of large BMD loss than BMD increasing with aging and, therefore, are beneficial to reduce the risk of bone disturbances since there is no consensus regarding positive changes with RT protocols, even in terms of small increments, on BMD.

## Figures and Tables

**Figure 1 healthcare-10-01129-f001:**
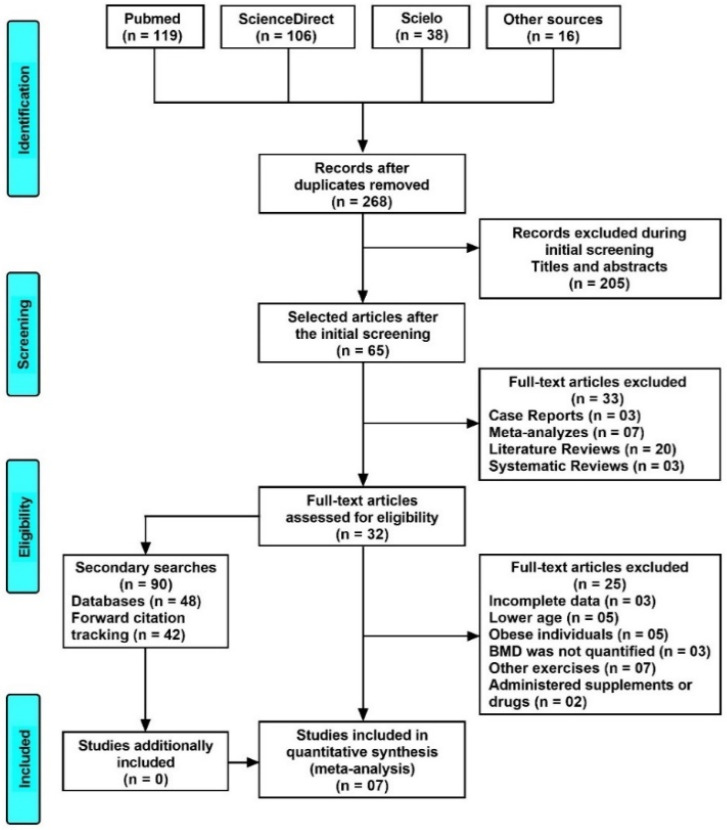
PRISMA flow diagram depicting the process of search and selection of the studies. BMD: bone mineral density.

**Figure 2 healthcare-10-01129-f002:**
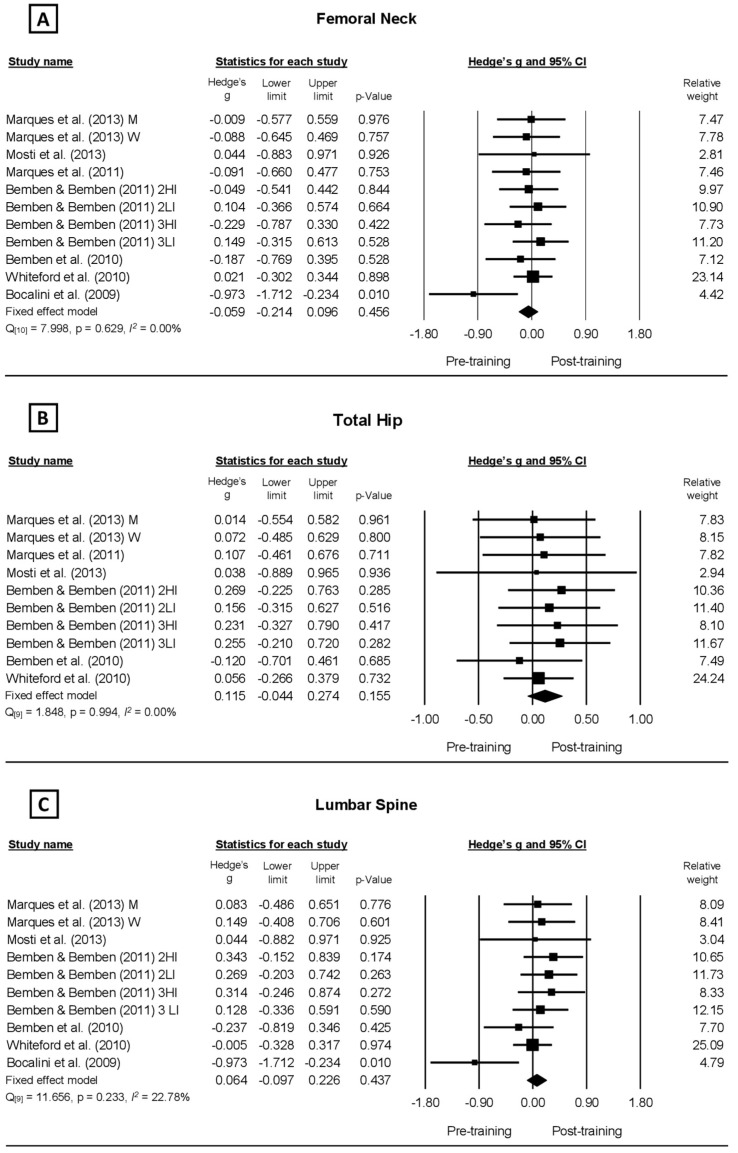
Forest plot analyzing the effect of resistance training protocols on bone mineral density reported by studies observing older populations. (**A**–**C**) illustrates the effect of the protocols on Femoral Neck, Total Hip, and Lumbar Spine, respectively.

**Figure 3 healthcare-10-01129-f003:**
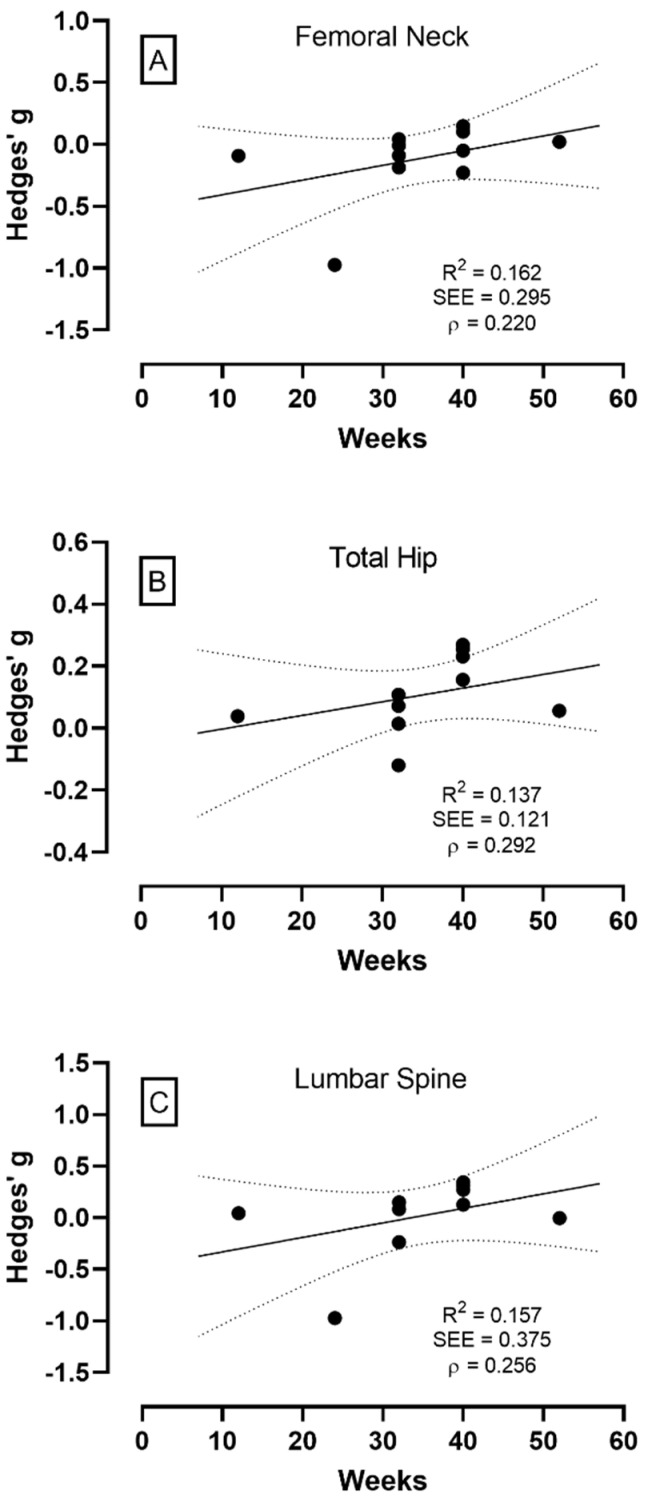
Regression analysis demonstrating the effect of the interventional period with resistance training on bone mineral density at femoral neck (panel **A**), total hip (panel **B**), and lumbar spine (panel **C**). The continuous line represents the regression trend, and the dashed line is the 95% CI.

## Data Availability

The data that support the findings of this study are available from the corresponding and last author (mario.espada@ese.ips.pt and dalton.pessoa-filho@unesp.br) upon reasonable request.

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
