# Peer review of "The Effect of Resistance Training on Bone Mineral Density in Older Adults: A Systematic Review and Meta-Analysis"

_healthcare, 2022, doi:10.3390/healthcare10061129_

Round 1
Reviewer 1 Report
The effect of resistance training on bone mineral density in elderly: A systematic review and meta-analysis
Summary
Seeks to summarize and determine the impact via effect sizes of resistance training for 12-52 weeks on BMD in older adults. The study followed PRISMA guidelines. The meta-analysis supports a preventive effect against reduced BMD with age.
Study Importance
The study provides further evidence on how important consistent resistance training can be for older adults/the aging adult to prevent natural BMD reductions.
General Comments
Older adult is considered the more proper term compared to elderly for those aged over 55 years.
Introduction: it would seem important to address how weightlifting supplies forces needed to effect BMD. What has been shown by previous studies (specific lifting movements, specific loading patterns) that demonstrate RT enhances bone health in older adults. Why did references 13 & 16 demonstrate an unsatisfactory effect? More details need to be introduced prior to leading into the final paragraph. What specifically are the discrepancies? Final paragraph suggests RT protocols but nothing prior suggests this.
Figure images a little grainy and makes them difficult to read at times. Recommend a better resolution.
More detail is needed in the table regarding each study. Stating the movements were just lower body, upper body, and trunk are too vague. What were the actual exercises performed? Recommend including the exercises in the table and an explanation of the exercise selection/how the exercise is performed in the discussion and whether the movements were load bearing specifically by the spine using free weights. Reference examples below.
Watson et al. (2017). High-Intensity Resistance and Impact Training Improves Bone Mineral Density and Physical Function in Postmenopausal Women With Osteopenia and Osteoporosis: The LIFTMOR Randomized Controlled Trial. Journal of Bone and Mineral Research.
Stengel et al. (2005) Power training is more effective than strength training for maintaining bone mineral density in postmenopausal women. Journal of Applied Physiology.
Stengel et al. (2007). Differential effects of strength versus power training on bone mineral density in postmenopausal women: a 2-year longitudinal study. British Journal of Sports Medicine.
Specific Comments
Line 27-28: PICO and PEDro not previously defined
Line 57-58: “values unchangeable” the wording here seems awkward. Suggest revising.
Line 63-66: Awkward wording within the sentences. Recommend revising.
Line 65: BMD sites – what are these sites? This needs to be addressed sooner in the intro. Also, BMD does not really have sites. References should be made the specific anatomical locations (bones, joint areas) where reduced BMD could be a problem.
Author Response
Dear reviewer,
I and my fellow authors would like to thank you for handling this manuscript, as well as for the comments and suggestions, which substantively enriched this manuscript.
We believe we have adequately addressed each of the comments. However, if you deem that more changes are necessary, we look forward to addressing any other concerns.
Thank you.
Best regards.

Reviewer 2 Report
Dear Authors,
It was a great pleasure to review your valuable work, and I hope that my suggestions and criticisms will help you to improve the quality of your publication.
I've included the review comments in the attached word doc.
Regards and thanks

Author Response

(The authors gave the same response as above.)

Round 2
Reviewer 1 Report
Thank you addressing the previous concerns.